# Skull-femoral traction followed by osteotomy correction is a safe and effective treatment for severe scoliosis with split cord malformation

**Mengyan Zhao[1], Fei Yuan[2], Chunjiang Lv[1], Xiaogang Shen[1], Hongzhe Qi[1], Gan Gao[2], Tao Guo[2]***

**1** Guizhou University of Traditional Chinese Medicine, Guiyang, China, **2** Guizhou Provincial People's Hospital, Guiyang, China

* gzy202406@163.com

**Data Availability Statement:** All relevant data are within the paper and its Supporting Information files.

## Abstract

### Objective

This study aimed to evaluate the safety and efficacy of skull-femoral traction followed by osteotomy correction in patients with severe spinal scoliosis and split cord malformation.

### Methods

We retrospectively analyzed ten cases of severe spinal scoliosis with Pang I type split cord malformation treated between August 2012 and August 2023. Patients underwent skull-femoral traction prior to osteotomy correction. We assessed changes in height, weight, coronal and sagittal Cobb's angles, and physiological indicators such as vital capacity (VC), forced vital capacity (FVC), forced expiratory volume in one second (FEV1), and blood gas levels before, during, and after treatment.

### Results

Traction duration ranged from 9 to 19 days, with height and weight showing significant increases post-treatment. The coronal Cobb's angle improved from pre-treatment to post-corrective surgery and remained stable at the final follow-up. Similar improvements were observed in the sagittal plane. Physiological indicators such as VC, FVC, and FEV1, as well as blood gas levels, normalized after treatment. Nutritional status, indicated by triceps skinfold thickness, albumin, and transferrin concentrations, also improved. No neurological complications or device-related complications occurred during or after treatment.

### Conclusion

Skull-femoral traction followed by osteotomy correction is a safe and effective treatment for severe spinal scoliosis with split cord malformation, offering an alternative to high-risk procedures.

**Funding:** National Natural Scientific Foundation of China, Grant/Award Number: 82260431; Basic plan project of Guizhou Provincial Science and Technology Department, Grant/Award Number: ZK [2022] Normal 247; Guizhou Province Science and Technology Support Plan, Guizhou Science and Technology Cooperation Support [2023] General 196; Guizhou Provincial People's Hospital, Grant/Award Number: GZSYBS [2021] 05 The funders had no role in study design, data collection and analysis, decision to publish, or preparation of the manuscript.

**Competing interests:** The authors have declared that no competing interests exist.

## Introduction

Severe scoliosis accompanied with split cord malformation, a rare congenital spinal cord malformation, is structurally complex, with high surgical difficulty and risk. It poses a challenge in the treatment of scoliosis, particularly in cases of Pang I-type split cord malformation, where the preventive excision of the septum before spinal correction is a focus of debate in spinal surgery [1–4]. Split cord malformation (SCM), also known as diastematomyelia, is an infrequent variant of a closed neural tube defect characterized by the longitudinal division of the spinal cord, resulting in the formation of duplicated neural tubes or two hemicords within a single dural sleeve. This condition typically arises from anomalies within the spinal canal and congenital spinal deformities during embryogenesis, and is frequently comorbid with congenital scoliosis (CS) [5]. Studies by McMaster [6] and Bradford [7] have respectively indicated a 16% and 17% incidence of diastematomyelia among individuals with CS. Shen [8] identified SCM as the most prevalent spinal intradural anomaly in the context of CS. The occurrence of CS in SCM patients is documented to range from 52% to 79% [9–12]. Select the treatment plan based on the benefits and risks of alternative options [13]. Longitudinal spinal traction has gained consensus for improving flexibility and reducing correction risk in severe scoliosis [14–17]. Some scholars have reported favorable correction outcomes with two-stage correction and internal fixation following heavy traction for severe scoliosis [18–25]. It's believed that skull-femoral traction can effectively prevent excessive correction angles during traction, thereby avoiding spinal cord and important neurovascular damage [26,27]. In our department, skull-femoral traction followed by correction and internal fixation has yielded good results, with 82 cases completed, including 10 cases of accompanying split cord malformation, all without irreversible nerve damage. Therefore, a retrospective analysis was conducted on cases of severe scoliosis with split cord type I malformation treated with skull-femoral traction followed by two-stage correction, aiming to explore the safety and clinical efficacy of this treatment approach.

## Data and methods

### 1. General information

Inclusion criteria: Cobb's angle ≥90° scoliosis accompanied by Pang I-type osseous split cord malformation. Exclusion criteria: (1) Patients with preoperative neurological symptoms; (2) Those with other complex intraspinal abnormalities such as tethered cord, tumors, or spinal meningocele; (3) Contraindications for skull-femoral traction such as skin infection in the operative area or cervical dislocation or instability. From August 2012 to August 2023, 10 cases of severe scoliosis accompanying with Pang I-type osseous split cord malformation were strictly selected from patients admitted to our department, including 4 males and 6 females, aged 13 to 34 years (mean age: 22.75 ± 6.19 years). All participants are fully informed of the relevant risks and benefits, and sign an informed consent form. All underage participants obtained the consent of their parents or guardians and signed an informed consent form. The individual in this manuscript has given written informed consent (as outlined in PLOS consent form) to publish these case details. The data was accessed for research purposes on May 1, 2023. The author has the right to obtain information that can identify individual participants during or after data collection. All patients underwent routine preoperative full-spine X-rays, CT, and MRI examinations. The main curve was located in the thoracic segment in 9 cases and thoracolumbar segment in 1 case. According to the classification of congenital scoliosis: 4 cases were segmental malformation, 3 cases were formation disorders, and 3 cases were mixed type; among them, 3 cases were complicated by syringomyelia. The preoperative main curve

Cobb's angle ranged from 62˚ to 176˚ (mean: 89.50 ± 23.13˚), bending Cobb's angle ranged from 59˚ to 172˚ (mean: 82.54 ± 15.35˚), flexibility ranged from 4.32% to 19.55% (mean: 11.36 ± 7.45%); no preoperative neurological abnormalities were found in all patients. The osseous split cord malformation was located at the apex of the main curve in 3 cases, at the far end of the apex vertebra in 6 cases (segments 1–3), and at the proximal end of the apex vertebra in 1 case.

## 2. Treatment methods

### 2.1 Skull-femoral traction

Skull-femoral traction is typically initiated on the second day following routine preoperative examinations. Initially, a 2 kg weight is applied, with gradual increments of 2 kg per day if well tolerated by the patient. The maximum traction weight may reach up to 50% of the patient's total body weight, contingent upon individual tolerance levels. Traction is administered for a minimum of 12 hours daily, with the weight reduced to 50% during the night. If tolerated, traction can be extended up to 20 hours per day. During non-traction periods, patients are allowed to leave traction for bathroom privileges, hygiene purposes, and eating. Throughout traction, frequent neurological assessments are conducted. Any observed hyperreflexia of the extremities, Babinski sign, paresthesia, dysfunction of cranial nerves, or other neurological compromises prompt immediate reduction in traction weight. In the event of complications related to the screw path, traction weight is decreased and maintained, with wound care performed until complete healing of the screw path, after which traction is resumed. The duration of traction is primarily determined by radiographic evidence of curve improvement on weekly radiographs, supplemented by clinical evaluations of pulmonary and neurological function.

**2.2 Scoliosis surgery.** The correction surgery began with the patient in a prone position, with the skull-femoral traction device still in place, and a "horseshoe-shaped" soft pad was used for positioning. A midline incision was made, followed by the dissection of subcutaneous tissue and subperiosteum from the spinous processes, laminae, and transverse processes. The muscles were dissected, and the laminae and articular processes within the fusion range were exposed adequately. The supra and interspinous ligaments at each level were completely removed. The inferior facet of the superior vertebra and much of the superior facet of the inferior vertebra were then removed. Pedicle screws were inserted into the vertebrae with the aid of 3D printed models (all patients used spinal internal fixation systems provided by Changzhou Kanghui). To prevent spinal cord compression or excessive accumulation causing spinal cord injury after correction, 2–3 vertebral plates were removed in the osteotomy area. For patients with large angles despite skull-femoral traction, pedicle subtraction osteotomy or asymmetric osteotomy of adjacent vertebrae was performed. For those with relatively small angles, only Smith-Petersen osteotomy or Ponte osteotomy was performed in the apex vertebra region. The number of osteotomies could be increased based on the correction needs. Before completing the osteotomy, a single-sided rod was placed, and the traction device was loosened to prevent displacement of the osteotomy ends compressing the dural sac. In all patients, the upper articular processes of the vertebrae in the fusion segment were removed to increase the correction effect and create a bone graft bed. After the correction, an anesthesia awakening test was performed. The laminae were roughened, and the fusion segment was adequately grafted with bone. Drains were placed, and the incisions were sutured layer by layer. After regaining consciousness from anesthesia, the patient was transferred to the recovery area.

### 3 Evaluation indicators

Measurements of height, weight, full-spine X-rays, and lung function tests (vital capacity, forced vital capacity, forced expiratory volume in one second, PO2, PCO2) were performed before skull-femoral traction, after traction, one week after correction surgery, and at the final follow-up. Skinfold thickness of the triceps, albumin, and transferrin were also evaluated.

### 4 Statistical methods

SPSS 22.0 software was used for statistical analysis. Paired sample t-test was used to analyze the evaluation indicator data before and after skull-femoral traction, postoperatively, and at the final follow-up. A P-value <0.05 was considered statistically significant.

## Results

### 1 Traction status and therapeutic effects

The traction duration for 10 patients ranged from 9 to 19 days (mean 12.13±4.66 days). Before traction, after traction, after corrective surgery, and at the last follow-up, the height, weight, and Cobb's angle were measured (Table 1). The height increased by an average of 4.75±3.12 cm, and the weight increased by 0.62±0.22 kg. The coronal and sagittal Cobb's angle correction rates after traction were 31.55±8.76% and 21.17±5.14%, respectively. All four parameters showed statistically significant differences compared to before traction (P<0.05). One patient experienced a decrease in muscle strength in the right lower limb after 1 weeks of traction. Traction was immediately released, and muscle strength was restored after close observation for three days. Subsequently, the patient underwent SPO osteotomy fusion. At the time of traction termination, the main curve Cobb's angle was 62˚. Another patient experienced numbness in the saddle area after 2 weeks of traction. Traction was immediately released, and gradual recovery occurred after seven days. The patient then underwent asymmetric shortening osteotomy of the adjacent vertebrae. At the time of traction termination, the main curve Cobb's angle was 57˚. None of the other patients experienced any neurological complications or complications related to the traction process, corrective surgery, or postoperative period. Complications such as loosening of the implants, infection, or nonunion of the incision were not observed during the traction process. Lung function improved to varying degrees in all 10 patients (Table 2). The improvement rates of VC, FVC, and FEV1 were 12.85±7.72%, 14.21±8.30%, and 6.44±2.78%, respectively. FEV1/FVC(%), P02, and PCO2 returned to normal range, and the differences were statistically significant compared to before traction (P<0.05). Nutritional status also improved to varying degrees (Table 3). The improvement rates of triceps skinfold thickness, albumin concentration, and transferrin concentration were 11.55±4.60%, 21.96±7.75%, and 23.13±8.51%, respectively, with statistically significant differences compared to before traction (P<0.05).

**Table 1. Height, weight and Cobb angle before traction, after traction, 1 week after operation and final follow-up.**

|  | Before traction | After traction | 1 week after operation | Final follow-up |
|---|---|---|---|---|
| Height(cm) | 153.13±9.28 | 157.88±9.66① | 169.64±9.87② | 169.38±9.95③ |
| Weight(kg) | 47.20±4.55 | 47.84±4.19① | 49.21±4.22② | 50.32±5.36③ |
| Cobb angle of coronal plane(˚) | 89.50±23.13 | 74.88±21.88① | 51.34±7.47② | 51.26±7.42③ |
| Cobb angle of sagittal plane(˚) | 76.68±13.72 | 62.47±12.46① | 38.51±5.83② | 38.64±5.87③ |

Note: ①Compared with before traction,P<0.05; ②Compared with before traction,P<0.05; ③Compared with before traction,P<0.05.

**Table 2. Pulmonary function before traction, after traction and 1 week after operation.**

| | Before traction | After traction | 1 week after operation |
|---|---|---|---|
| Vital capacity(L) | 3.75±0.26 | 4.20±0.04① | 4.22±0.05② |
| Forced vital capacity(L) | 3.65±0.26 | 4.14±0.04① | 4.16±0.04② |
| Forced expiratory volume in one second(L) | 3.34±0.22 | 3.54±0.15① | 3.54±0.15② |
| FEV1/FVC(%) | 0.92±0.01 | 0.86±0.03① | 0.84±0.04② |
| $PO_2$(mmHg) | 59.50±5.25 | 70.88±3.59① | 71.45±3.64② |
| $PCO_2$(mmHg) | 40.63±1.63 | 41.88±1.84① | 41.76±1.67② |

Note: ①Compared with before traction,$P<0.05$; ②Compared with before traction,$P<0.05$.

**1.1 Perioperative and follow-up details.** The surgical duration for the 10 patients ranged from 230 to 410 minutes, with an average of 305.32±78.76 minutes. Intraoperative blood loss ranged from 660 to 1800 ml, with an average of 854.76±443.55 ml. No operation was performed on the split cord malformation, and five cases underwent single-segment pedicle subtraction osteotomy, four cases underwent asymmetric shortening osteotomy of adjacent vertebrae in two segments, and one case underwent in-situ fusion. Cobb's angle of the main curve in the coronal and sagittal planes before traction, after traction, after corrective surgery, and at the last follow-up were measured (Table 1). The correction rates in the coronal and sagittal planes at the last follow-up were 53.54±5.17% and 52.55±4.69%, respectively, which showed statistically significant differences compared to before traction (P<0.05). No neurological injuries, major vascular damage, cerebrospinal fluid leakage, infections, or deaths occurred during or after the surgery. Lung function and nutritional status significantly improved (Table 2). The improvement rates of VC, FVC, and FEV1 were 14.21±7.81%, 15.34 ±8.56%, and 6.68±2.79%, respectively. FEV1/FVC(%), P02, and PCO2 returned to normal range, and the differences were statistically significant compared to before traction (P<0.05). Nutritional status also improved to varying degrees (Table 3). Triceps skinfold thickness, albumin concentration, and transferrin concentration improved by 14.12±4.97%, 23.12±7.87%, and 25.43±8.18%, respectively, with statistically significant differences compared to before traction (P<0.05). The follow-up period ranged from 6 to 36 months. None of the 10 patients experienced any neurological functional deterioration. At the last follow-up, no cases of internal fixation displacement, loosening, or fracture were observed. The patients' general condition improved, and their appearance and trunk balance significantly improved compared to before surgery.

## Discussion

Spinal cord split is divided into three types [28,29]: Type I is a double-tube type, where both halves of the spinal cord have their own independent dural tubes, separated by a bony or cartilaginous septum. For patients with congenital spinal deformities associated with spinal cord

**Table 3. Nutritional status before traction, after traction and 1 week after operation.**

| | Before traction | After traction | 1 week after operation |
|---|---|---|---|
| Triceps skinfold thickness(mm) | 11.36±3.21 | 12.46±3.06① | 14.53±3.64② |
| Albumin(g/L) | 30.95±1.98 | 39.66±2.33① | 40.27±3.36② |
| Transferrin(g/L) | 1.96±0.11 | 2.55±0.10① | 2.63±0.11② |

Note: ①Compared with before traction,$P<0.05$; ②Compared with before traction,$P<0.05$.

split without neurological symptoms or with stable neurological symptoms, especially for Type I spinal cord split patients, there is still controversy regarding the need for prophylactic resection of the spinal cord split before spinal correction surgery. Some scholars believe that significant changes occur in the morphology of the spinal canal during the reconstruction of the coronal and sagittal plane, causing a certain amount of traction and compression on the spinal cord, thereby exacerbating existing nerve damage or causing new nerve damage. They suggest that all patients with bony spinal cord split should undergo prophylactic laminectomy before spinal correction surgery [30]. However, Miller et al. [31] reported that among 33 patients who underwent laminectomy for spinal cord split, 22 patients did not show significant improvement in postoperative neurological symptoms, and one patient experienced worsened neurological symptoms. They believe that surgical intervention for spinal cord split patients should be approached with extreme caution. Chen et al. [4] also found that the incidence of complications in patients with congenital spinal deformities associated with spinal cord split who underwent one-stage posterior spinal osteotomy correction surgery was not significantly higher compared to patients without spinal cord split. Shen et al.'s [32] prospective study results indicated that for patients with spinal cord split who have no neurological symptoms or stable neurological symptoms, a simple spinal correction surgery is safe and effective without the need for prophylactic laminectomy. Zhu et al. [33] suggested that for patients with severe spinal deformities and Type I spinal cord split who require three-column osteotomy during surgery, if the bony split is within the osteotomy level, it is advisable to simultaneously remove the lamina during the correction surgery. The viewpoints of the aforementioned authors are based on one-stage surgery and do not consider the impact of traction on spinal cord split. Multiple perspectives suggest that longitudinal spinal traction has a positive effect on improving flexibility and reducing the risk of correction in severe rigid spinal scoliosis, but there is no clear study on the impact of traction therapy on spinal cord split.

This article aims to analyze the safety and effectiveness of cranial traction on severe spinal scoliosis accompanied by spinal cord bony split. Based on clinical practice, the author summarizes the following points after performing cranial femoral traction followed by one-stage surgery on suitable cases: 1. Cranial femoral traction has strong stability, with continuous and stable traction force. Its effect on longitudinal spinal traction can be seen as a tearing-repair reconstruction effect on contracted soft tissues. Observations on patient's skin appearance, skin tension, and muscle tension lead to the conclusion that it has similar effects on spinal intervertebral discs, facet joints, ligaments, and other tissues. 2. Cranial femoral traction has good consistency in traction force direction, restoring coronal and sagittal plane balance of the spine. Observations of severe spinal scoliosis patients after traction lead to this conclusion. 3. Post-cranial femoral traction increases thoracic cavity volume, improves lung function indicators to varying degrees, alleviates abdominal organ compression, etc., leading to improvements in patient nutritional status, enhancing perioperative safety, and surgical tolerance. Mesenteric vessels, mesenteric ligaments, etc., undergo reconstruction during traction, reducing postoperative complication risks. 4. Post-cranial femoral traction reduces the angle of lateral scoliosis, significantly influencing surgical decisions. The main curve, compensatory curve, scoliosis, and rotation all show varying degrees of improvement, with some correction rates reaching 50%. Compared to one-stage correction, the decision-making in traction surgery regarding osteotomy level, osteotomy range, fusion range all decreases, reducing intraoperative correction difficulty, surgical time, and blood loss significantly. 5. Compared to gravity traction and halo pelvic traction, cranial femoral traction is stable, continuous, and exerts strong traction force, allowing patients to choose rest, walk freely, and take care of themselves, thus enhancing their dignity. 6. Cranial femoral traction can increase the tolerance of the spinal cord to deformation; some patients experience an 8 cm increase in height after traction, while the total

length of the spinal canal remains unchanged due to changes in spinal cord morphology with the spinal canal. Blood vessels and spinal cord adapt during slow deformation, stabilizing in the final traction state. Cranial femoral traction guides the degree of correction in surgery. 7. For those with abnormalities in the spinal canal or spinal cord, progressive traction allows observation of neurological function status. If any neurological abnormalities appear, traction is immediately relaxed, leading to neurological function recovery, indicating the limit of correction and avoiding over-correction in surgery. Currently used traction methods include Halo-pelvic traction, Halo-gravity traction, and Halo-femoral trochanter traction. Qiu et al. [34,35] reported that Halo wheelchair suspension gravity traction can improve the correction effect of spinal scoliosis, especially idiopathic scoliosis. However, gravity traction requires a longer duration, with reported maximum traction times in China ranging from 4 to 15 weeks, averaging 10.4 weeks, leading to prolonged hospital stays. Moreover, the stability of such traction is poor, and its actual correction ability requires further observation. Halo-femoral trochanter traction [36]: (1) Heavy-duty Halo-femoral trochanter traction provides greater traction force, allowing sufficient correction force for deformed spines, effectively releasing the apex vertebra, which is difficult to achieve with other methods; (2) Heavy-duty Halo-femoral trochanter traction allows patients to have more effective traction time, maintaining traction for a long time, especially during nighttime sleep when muscles are relatively relaxed, resulting in better traction effects; (3) Heavy-duty Halo-femoral trochanter traction has a better loosening effect on rigid spinal scoliosis, particularly within the main curve region where true stiffness exists, while adjacent segments above and below the main curve are relatively stiff; Due to the large traction force and long duration of heavy-duty Halo-femoral trochanter traction, it can loosen the apex vertebra region within the main curve area, adjacent segments above and below the main curve, and each intervertebral space, releasing more mobility, effectively loosening contracted soft tissues in the main curve area including trunk muscles, paraspinal muscles, etc. Halo-femoral trochanter traction can reduce the incidence of intraoperative spinal cord neurological complications, but the traction process itself may lead to a series of neurological damage complications. Bilateral Halo-femoral trochanter traction increases bedridden time, leading to long-term bed-related complications such as pressure sores, dependent lung infections, lower limb deep vein thrombosis, etc., and can also cause nail bed infections or intracranial infections; Due to prolonged traction, joint stiffness and other bone and joint complications may occur, such as knee joint stiffness, hip joint stiffness, etc. These characteristics are similar to cranial femoral traction.

The article mainly discusses the safety and clinical efficacy of cranial traction in the treatment of severe spinal scoliosis with Type I bone-type spinal cord longitudinal fissure. In typical cases (Figs 1–4), neurological dysfunction was restored after retreatment, suggesting non-organic damage. The specific reasons are evidenced by spinal cord MRI images, where the dual canal spinal cord is close to the concave side vertebral canal wall before traction. After traction, the angle of lateral convexity decreases, and the dual canal spinal cord gradually passively drifts towards the convex side. The concave side spinal cord experiences neurological dysfunction due to bone fissure compression at the "critical point". The compression is relieved after retreatment, leading to neurological recovery. In a typical case 1, the bone fissure is located at the top vertebra. During the surgery, bone fissure excision and invasive vertebral canal osteotomy were not performed. In-situ fixation and fusion were conducted, resulting in no postoperative neurological dysfunction. In a typical case 2, the bone fissure is located in the lower three segments of the top vertebra within the conus medullaris area. The conus medullaris is divided into two, and an asymmetric shortening osteotomy of the top vertebra is performed. No postoperative neurological dysfunction was observed. Among the 10 patients with Type I bone-type spinal cord longitudinal fissure, none underwent bone lamina excision

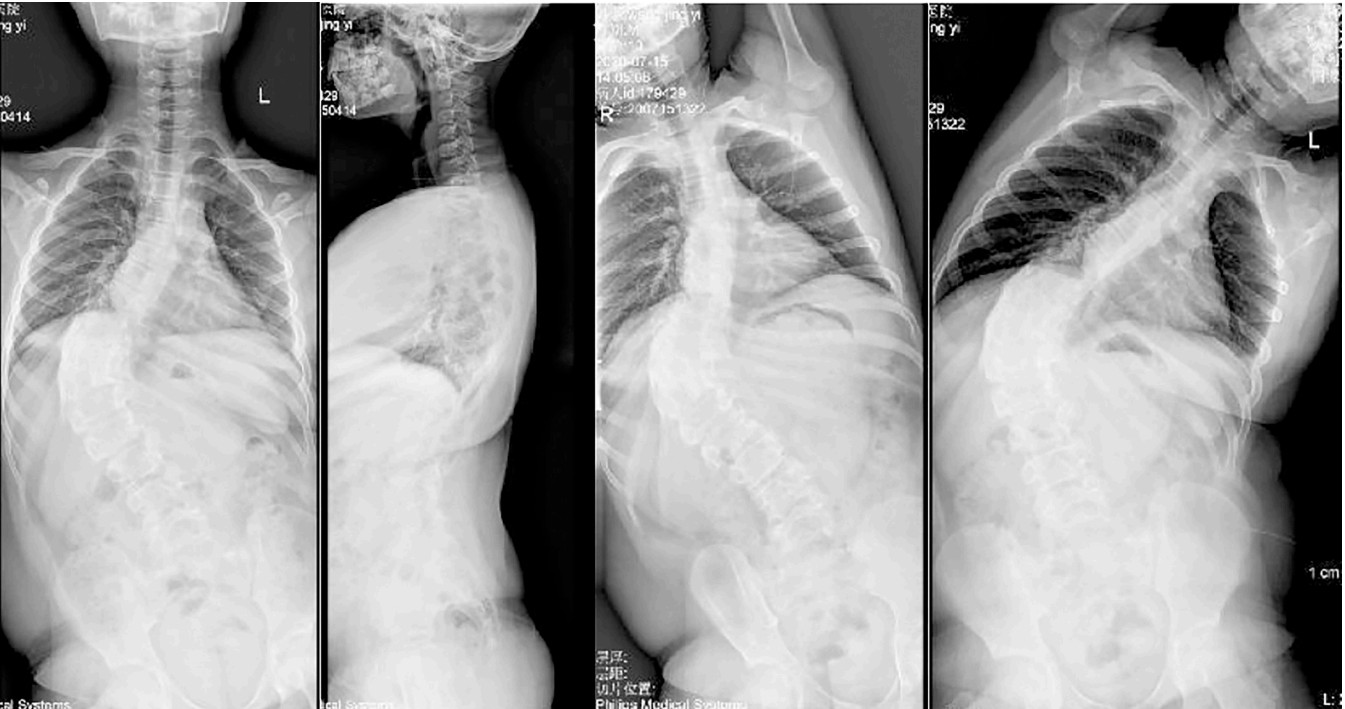

**Fig 1. Preoperative X-ray in anteroposterior and lateral, and bending positions.**

surgery. None of them experienced neurological decline after corrective surgery, and there was no occurrence of internal fixation displacement, loosening, or fracture at the last follow-up. The patients generally had good overall condition with improved appearance and trunk balance compared to before the surgery, as well as a noticeable improvement in nutritional status. Cranial traction has the advantages mentioned above and is considered safe and effective. However, it is not the sole option for patients with spinal deformities and has several drawbacks, such as a higher risk of nail-related complications, potential paralysis risks, limited tolerance of the cervical spine, risks of cranial nerve and brachial plexus traction, risks of osteoporosis, patient psychological burden, and nursing difficulties. In this study, we enrolled 10 participants, a number that indeed falls short of the ideal sample size. This limitation was primarily due to the availability of the specific population. Nonetheless, our data analysis revealed statistically significant results that are consistent with findings in the existing literature. The limitation in sample size may affect the generalizability of the results. To ensure the reliability of our findings, we implemented rigorous data quality control measures. To further validate our results, we have planned follow-up studies that will include a broader range of participants and a larger sample size. We believe this will help to strengthen the generalizability and applicability of our conclusions.

While the present study has focused on the immediate safety and efficacy of cranial traction in the treatment of severe spinal scoliosis with Type I bone-type spinal cord longitudinal fissure, it is imperative to consider the long-term implications of this treatment modality. Long-term follow-up is essential to monitor the durability of the surgical correction, the progression of any underlying spinal deformities, and the potential emergence of late-onset complications. One of the primary concerns in the long-term management of patients with spinal cord split and severe spinal deformities is the potential for neurological complications. These may include the development of new neurological deficits or the exacerbation of pre-existing

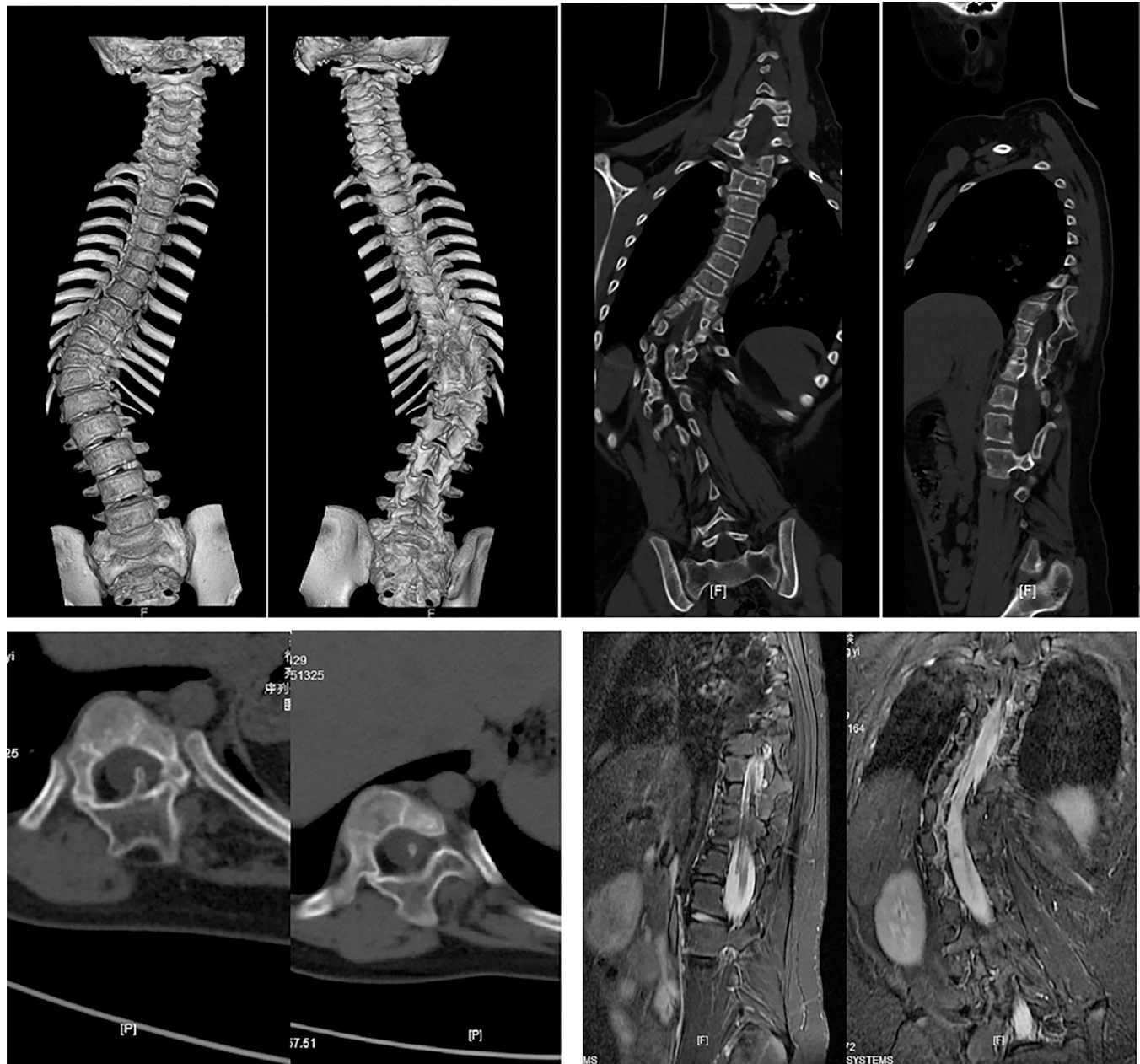

**Fig 2. Preoperative CT three-dimensional reconstruction, coronal reconstruction, axial reconstruction, and MRI examination.**

conditions due to the stresses placed on the spinal cord during and after surgery. It is crucial to establish protocols for regular neurological assessments to detect any subtle changes that may not be immediately apparent postoperatively. Additionally, the long-term impact of spinal correction on the overall health and quality of life of patients must be evaluated. This includes the potential for chronic pain, the need for subsequent surgeries, and the psychological impact of living with a corrected but potentially altered spinal structure. The incidence of secondary conditions such as osteoporosis, which may be exacerbated by long-term immobilization and surgical intervention, should also be monitored. Furthermore, the study acknowledges the

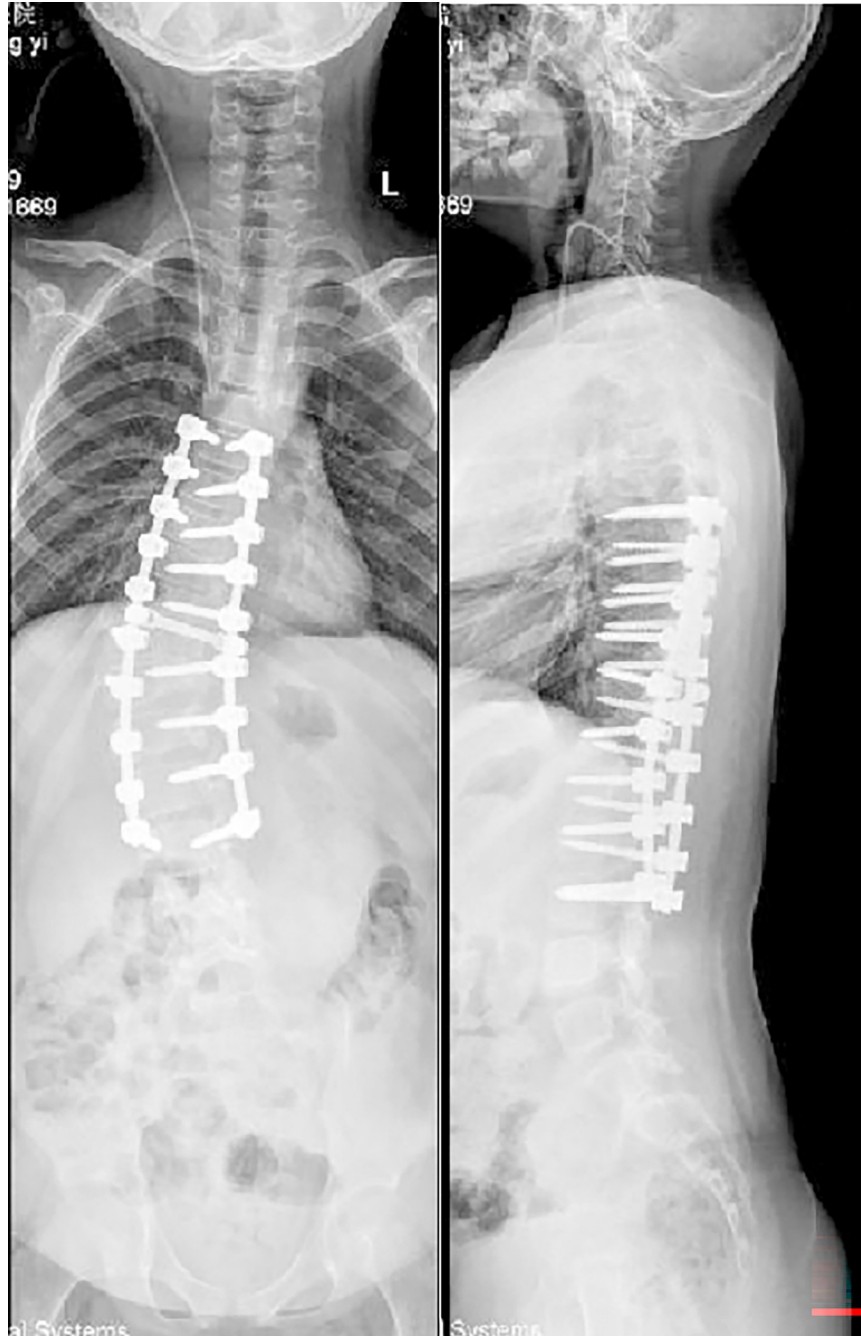

**Fig 3. Postoperative X-ray in anteroposterior and lateral.**

limitations in sample size, which may affect the generalizability of the findings. Future research with a larger cohort will be instrumental in providing more robust data on the long-term outcomes of cranial traction in patients with spinal cord split. It will also be important to investigate the impact of different traction methods on the spinal cord and surrounding tissues over time.

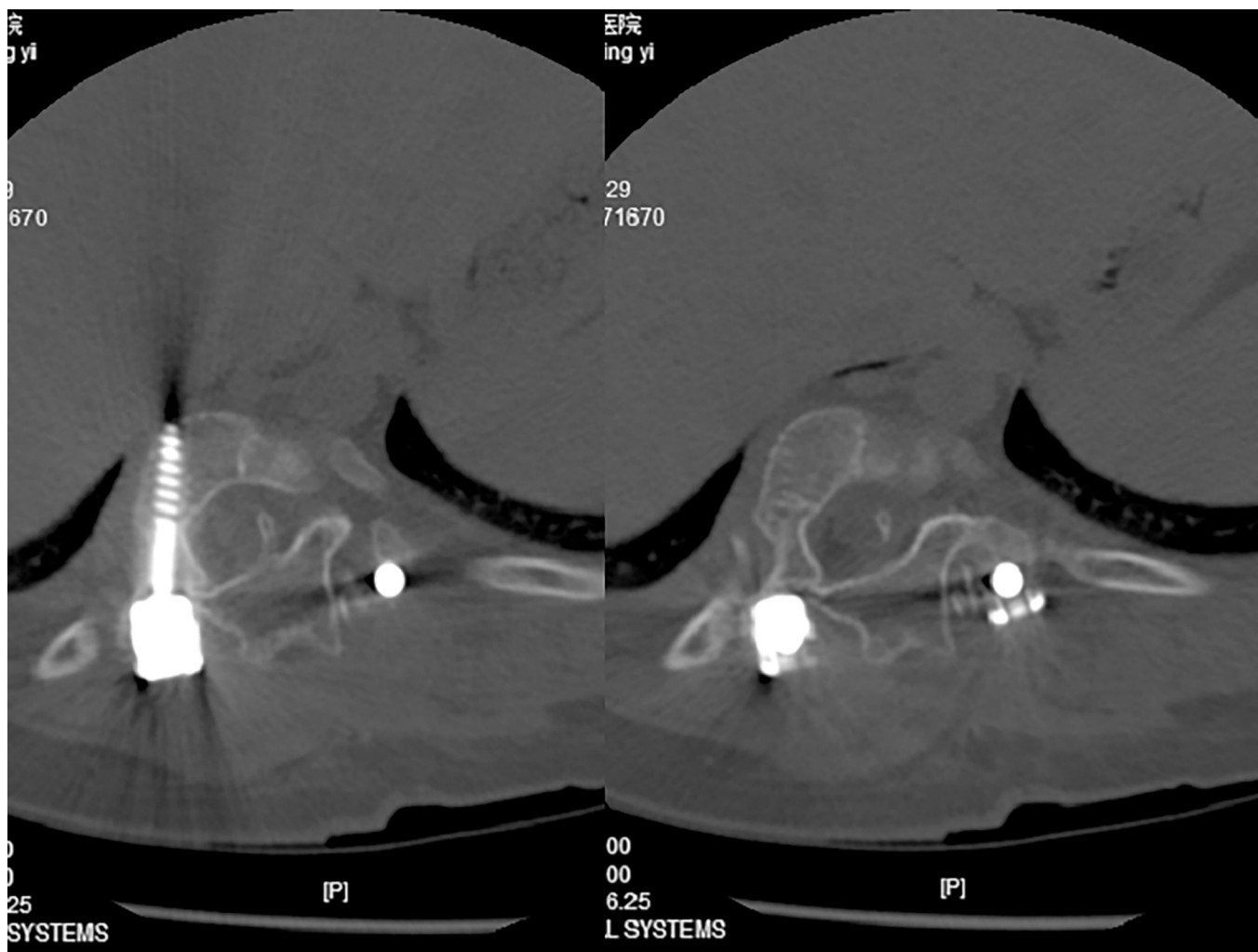

**Fig 4. Postoperative CT scan in the axial plane.**

## Conclusion

In summary, cranial traction is a safe and effective treatment method for severe spinal scoliosis accompanied by spinal cord bone-type longitudinal fissure. It can avoid the higher risk of bone lamina excision surgery and is considered one of the preferred treatment options. However, further clinical verification is still needed.

## Supporting information

**S1 Raw data. The raw data tables can be found in the supporting information.** (XLSX)

## Author Contributions

**Conceptualization:** Mengyan Zhao, Gan Gao, Tao Guo.

**Data curation:** Mengyan Zhao, Fei Yuan, Chunjiang Lv, Xiaogang Shen, Hongzhe Qi, Gan Gao.

**Formal analysis:** Mengyan Zhao, Fei Yuan, Chunjiang Lv, Xiaogang Shen, Hongzhe Qi, Tao Guo.

**Investigation:** Fei Yuan.

**Methodology:** Mengyan Zhao, Fei Yuan, Chunjiang Lv, Xiaogang Shen.

**Validation:** Hongzhe Qi.

**Writing – original draft:** Mengyan Zhao.

**Writing – review & editing:** Tao Guo.

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
