## [Decision Letter · Decision Letter 0]

31 Jul 2024

PONE-D-24-23086Skull-femoral traction followed by osteotomy correction is a safe and effective treatment for severe scoliosis with split cord malformationPLOS ONE

Dear Dr. Guo,

Thank you for submitting your manuscript to PLOS ONE. After careful consideration, we feel that it has merit but does not fully meet PLOS ONE’s publication criteria as it currently stands. Therefore, we invite you to submit a revised version of the manuscript that addresses the points raised during the review process.

We look forward to receiving your revised manuscript.

Kind regards,

Barry Kweh

Academic Editor

PLOS ONE

2. We note that your Data Availability Statement is currently as follows: "All relevant data are within the manuscript and its Supporting Information files."

5. We note that Figure "Typical Case E" includes an image of a patient or participant in the study.

Reviewer's Responses to Questions

**Comments to the Author**

1. Is the manuscript technically sound, and do the data support the conclusions?

Reviewer #1: Partly

Reviewer #2: Yes

Reviewer #3: Partly

2. Has the statistical analysis been performed appropriately and rigorously? 

Reviewer #1: Yes

Reviewer #2: Yes

Reviewer #3: Yes

3. Have the authors made all data underlying the findings in their manuscript fully available?

Reviewer #1: Yes

Reviewer #2: Yes

Reviewer #3: Yes

4. Is the manuscript presented in an intelligible fashion and written in standard English?

Reviewer #1: Yes

Reviewer #2: Yes

Reviewer #3: Yes

5. Review Comments to the Author

Reviewer #1: Dear Authors, I had the opportunity to review your article, which addresses a significant topic in orthopedic and spinal surgery: the treatment of severe scoliosis with split spinal cord malformation using craniofemoral traction followed by osteotomy. The topic is relevant and well-chosen, which is why certain sections require substantial elaborations. The introductory section is rather sparse and does not provide adequate context regarding the prevalence of the condition, the main clinical challenges, and the currently available treatment options. It would be beneficial to include a more detailed literature review that explores the effectiveness of alternative treatments and better positions craniofemoral traction within the landscape of therapeutic options. The methods section is generally clear but lacks specific details crucial for the replicability of the study, particularly the inclusion and exclusion criteria. What specific criteria were used to select the 10 patients? Were there any excluded patients, and if so, for what reasons? The description of the traction technique lacks details about the precise protocol followed (e.g., the duration of daily traction sessions, the forces applied, etc.), as well as the potential risks and benefits. The section does not specify whether there were standardized postoperative rehabilitation measures for all patients. This implies that the physician must necessarily provide the patient with all appropriate information about the potential risks, benefits, and specific advantages of one procedure versus another. In this context, I suggest including an article that I recommend citing in your references (https://doi.org/10.1016/j.jflm.2024.102674), which clearly explains that in carrying out a thorough informed consent process, physicians have a professional duty to outline and initiate a discussion about the risks, benefits, and possible alternatives to a given procedure. The results presented are interesting and show significant improvements in various parameters. However, there is a lack of detailed discussion of the statistical analyses used. A more extensive discussion of long-term follow-up data would be useful. For example, how much time was considered in the last follow-up? Were there any cases of scoliosis recurrence or late complications? The data presented are averages, but there is no mention of individual variability among patients. Did some patients respond better to the treatment than others? If so, what factors might explain this variability? The discussion of the results is adequate but could be enriched in several ways. For instance, the section could benefit from a more in-depth explanation of the physiological mechanisms underlying the observed improvements in patients, particularly regarding the improvement in pulmonary function and nutritional status. Additionally, it is important for the authors to acknowledge and discuss the limitations of their study. For example, the sample size is rather small (10 patients), which may limit the generalizability of the results. Moreover, the lack of a control group limits the ability to definitively attribute the observed effects to the specific treatment. In conclusion, the article presents promising results regarding the use of craniofemoral traction followed by osteotomy for the treatment of severe scoliosis with split spinal cord malformation. However, to enhance the scientific and clinical value of the work, more thorough methodological elaboration, a more critical discussion, and an acknowledgment of the study's limitations are necessary. Expanding these sections would significantly improve the robustness and impact of the article.

Reviewer #2: 1. This article is a retrospective study. Only 10 patients were included. Using self-control, whether the number of cases is too small.

2. The surgical methods used by these 10 patients are different. Whether there is a difference in the base number between different surgical methods and whether it has an impact on the postoperative data. If there is any relevant data, please add it.

Reviewer #3: Evaluation of Research Innovation

1. Innovative Aspects:

Novel Treatment Method**: The study employs a novel approach of skull-femoral traction combined with osteotomy correction for treating severe spinal scoliosis accompanied by split cord malformation, which represents a relatively new method in this treatment area.

Comprehensive Assessment Metrics**: The research not only focuses on correcting spinal angles but also encompasses physical indicators such as height, weight, and physiological functions like pulmonary function and nutritional status, which is less commonly seen in similar studies.

2. Research Contributions:

Filling Clinical Practice Gaps: The study addresses gaps in treatment methods for severe spinal scoliosis accompanied by split cord malformation, particularly by avoiding potentially higher-risk procedures like spinous process resection.

Validation of Method Safety and Efficacy: Results indicate that skull-femoral traction followed by osteotomy correction was safe without neurological impairments or major complications during and after treatment. It effectively improves spinal deformities and physiological parameters, offering significant guidance for clinical practice in this patient group.

3. Suggestions for Improvement:

Long-term Follow-up Data: To further validate the method's long-term effects and stability, it is recommended to include extended follow-up data, particularly beyond one year post-treatment.

Exploration of Additional Physiological Parameters: In addition to the assessed metrics, consider adding other physiological parameters such as further analysis of cardiopulmonary function or assessment of patient quality of life to comprehensively evaluate treatment outcomes.

In summary, this study demonstrates innovation in method selection and comprehensive assessment, with clear potential contributions to clinical practice. However, there is room for further optimization and enhancement to improve the study's comprehensiveness and scientific rigor.

Abstract

Areas for Improvement:

Issue: The abstract is detailed but lacks conciseness, resulting in lengthiness that may overwhelm readers seeking a quick overview.

Recommendation: Simplify the presentation by focusing on essential findings and consider omitting specific numerical data that can be detailed in the main text. Enhance readability by streamlining the language.

Addressing these aspects will enhance the abstract's clarity, accessibility, and effectiveness in conveying the study's significance and findings within the orthopedic research community.

Introduction

The introduction to this study excels in describing background and conducting a literature review. However, enhancing methodological comparisons and future research prospects could further elevate its scientific value and academic contribution.

Areas for Improvement:

While the introduction provides comprehensive background and literature support, it could further emphasize the limitations of current treatment methods and potential directions for future research to enhance the optimization and innovation of treatment approaches.

It is recommended to include comparative analyses of other treatment methods, particularly effectiveness and pros and cons compared to skull-femoral traction followed by osteotomy correction, to enhance the study's academic depth and practical relevance.

Evaluation of Research Methodology

1. Methodological Strengths:

Comprehensive Approach: The study employs a comprehensive methodology that includes detailed pre- and post-treatment assessments of various parameters such as spinal angles, physiological measurements (e.g., VC, FVC, FEV1), and biochemical markers (e.g., albumin, transferrin). This approach provides a thorough evaluation of the treatment's effects on both structural and functional outcomes.

Longitudinal Assessment: The use of multiple follow-up points (after traction, after corrective surgery, and at the last follow-up) enhances the reliability of the findings by capturing changes over time and assessing the sustainability of treatment outcomes.

Absence of Complications: The study reports no neurological impairments or significant complications related to the traction or osteotomy procedures, indicating careful procedural management and patient monitoring.

2. Areas for Improvement:

Sample Size and Diversity: The study's sample size of ten patients limits the generalizability of findings. Including a larger and more diverse patient cohort could strengthen the study's external validity and allow for subgroup analyses based on different patient characteristics.

Control Group or Comparative Analysis: Introducing a control group or comparative analysis with alternative treatment methods would provide a clearer basis for evaluating the specific benefits of skull-femoral traction followed by osteotomy correction compared to standard treatments.

Data Standardization: Ensure consistent data collection protocols across all measurements to minimize variability and enhance the reliability of comparisons over time.

3. Data Analysis:

Statistical Rigor: The statistical analysis appears appropriate for the study's objectives, utilizing descriptive statistics and measures of central tendency to summarize continuous data. However, considering the complexity of the data (e.g., longitudinal measurements), employing more sophisticated statistical methods such as repeated measures ANOVA or mixed-effects models could provide deeper insights into treatment effects over time.

In conclusion, while the study demonstrates methodological rigor in its approach to evaluating skull-femoral traction followed by osteotomy correction for severe spinal scoliosis with split cord malformation, there are opportunities for enhancement, particularly in sample size, comparative analysis, and data standardization, to further strengthen the validity and generalizability of its findings.

Results

Areas for Improvement:

1. Clarity and Logic of Data Presentation:

Issue: Some sections of data presentation may be too compact, making it challenging for readers to follow, especially in describing different patient scenarios.

Recommendation: Consider reorganizing the data, such as through clearer segmentation to distinguish between different cases and treatment effects. Ensure each case description is clear and structured to facilitate reader comprehension of the study's progression.

2. Statistical Analysis and Significance:

Issue: Descriptions regarding statistical significance should be more detailed and thorough to ensure readers have a clear understanding of the reliability of the results.

Recommendation: Clearly state the methods of statistical analysis and significance levels (e.g., P-values) in each results subsection, and discuss their implications for interpreting the study findings.

3. Use of Figures and Tables:

Issue: The results section could benefit from additional figures and tables to enhance the visual representation of data, aiding readers in quickly and intuitively grasping key findings.

Recommendation: Consider adding relevant figures and tables to the results section to showcase primary data and trends. This will highlight key results and improvement trends from the study.

Discussion

Clarity of Logic and Structure:

Issue: The viewpoints and research findings in the discussion could be more clearly organized and presented, especially in describing different cases and treatment outcomes.

Recommendation: Consider reorganizing the discussion content to ensure there is a clear logical flow between each viewpoint and research result. Enhancing overall coherence with clearer paragraph structures and topic sentences would facilitate easier understanding and follow-up of the discussion's progression.

Conclusion

Please include a conclusion section at the end of the paper to match the abstract.

6. PLOS authors have the option to publish the peer review history of their article (what does this mean?). If published, this will include your full peer review and any attached files.

Reviewer #1: **Yes: **Giuseppe Basile

Reviewer #2: No

Reviewer #3: No

---

## [Author Response · Author response to Decision Letter 0]

7 Aug 2024

Response Letter to the Reviewers

Thank you very much for giving us the opportunity to submit a revised manuscript. We appreciate the reviewers for your constructive comments and valuable suggestions on our manuscript entitled “Skull-femoral traction followed by osteotomy correction is a safe and effective treatment for severe scoliosis with split cord malformation” (Manuscript ID: PONE-D-24-23086). 

We have carefully studied the valuable comments and tried our best to revise the manuscript. All revised portions are red-marked in the revised manuscript and supporting information. 

We have provided a point-by-point response to the reviewer’s comments below. Our responses to the reviewer’s comments are marked in blue, and the revisions in the manuscript and supporting information are marked in red below.

We would like to thank you once again for your work on our manuscript. If there are any questions, please do not hesitate to contact us.

Reviewer: 1

Dear Authors, I had the opportunity to review your article, which addresses a significant topic in orthopedic and spinal surgery: the treatment of severe scoliosis with split spinal cord malformation using craniofemoral traction followed by osteotomy. The topic is relevant and well-chosen, which is why certain sections require substantial elaborations. The introductory section is rather sparse and does not provide adequate context regarding the prevalence of the condition, the main clinical challenges, and the currently available treatment options. It would be beneficial to include a more detailed literature review that explores the effectiveness of alternative treatments and better positions craniofemoral traction within the landscape of therapeutic options. The methods section is generally clear but lacks specific details crucial for the replicability of the study, particularly the inclusion and exclusion criteria. What specific criteria were used to select the 10 patients? Were there any excluded patients, and if so, for what reasons? The description of the traction technique lacks details about the precise protocol followed (e.g., the duration of daily traction sessions, the forces applied, etc.), as well as the potential risks and benefits. The section does not specify whether there were standardized postoperative rehabilitation measures for all patients. This implies that the physician must necessarily provide the patient with all appropriate information about the potential risks, benefits, and specific advantages of one procedure versus another. In this context, I suggest including an article that I recommend citing in your references (https://doi.org/10.1016/j.jflm.2024.102674), which clearly explains that in carrying out a thorough informed consent process, physicians have a professional duty to outline and initiate a discussion about the risks, benefits, and possible alternatives to a given procedure. The results presented are interesting and show significant improvements in various parameters. However, there is a lack of detailed discussion of the statistical analyses used. A more extensive discussion of long-term follow-up data would be useful. For example, how much time was considered in the last follow-up? Were there any cases of scoliosis recurrence or late complications? The data presented are averages, but there is no mention of individual variability among patients. Did some patients respond better to the treatment than others? If so, what factors might explain this variability? The discussion of the results is adequate but could be enriched in several ways. For instance, the section could benefit from a more in-depth explanation of the physiological mechanisms underlying the observed improvements in patients, particularly regarding the improvement in pulmonary function and nutritional status. Additionally, it is important for the authors to acknowledge and discuss the limitations of their study. For example, the sample size is rather small (10 patients), which may limit the generalizability of the results. Moreover, the lack of a control group limits the ability to definitively attribute the observed effects to the specific treatment. In conclusion, the article presents promising results regarding the use of craniofemoral traction followed by osteotomy for the treatment of severe scoliosis with split spinal cord malformation. However, to enhance the scientific and clinical value of the work, more thorough methodological elaboration, a more critical discussion, and an acknowledgment of the study's limitations are necessary. Expanding these sections would significantly improve the robustness and impact of the article.

Response: 

1.Thank you for the valuable feedback from the reviewers. We have expanded the introduction section to include epidemiological context, clinical challenges, treatment options, a discussion on the risks and benefits of alternative treatments, and a comprehensive review of the relevant literature.

2.The reason for selecting the 10 patients in this study is that they met the theme of our study from our case repository; they all had congenital scoliosis complicated by spinal bifida occulta and underwent traction treatment and surgical treatment. These were the criteria for inclusion. We established exclusion criteria to filter out cases that did not fit the theme of the study. Four patients were screened out by the exclusion criteria, and due to word limitations, the reasons for each exclusion were not detailed, which does not affect the theme of the study.

3.We have provided a detailed description of the traction method in the Methods section. We have added references to the relevant literature.

4.In this study, the ten cases observed did not exhibit any long-term complications, such as worsening scoliosis, failure of internal fixation, paralysis, etc. Currently, none of the cases have been lost to follow-up, and no long-term complications have been reported. If continued follow-up does not have a designated endpoint, this study would be unable to conclude, and we can only set interim endpoints and report the findings. Therefore, this manuscript is presented in such a format. We hope to continue following up with the patients and report the observations in the future.

5.Due to word count limitations, we are unable to discuss each case individually. Since all patients have shown significant improvements across all observed indicators, they can be described collectively. Additionally, we have chosen to present typical cases as illustrative examples.

6.All patients surveyed did not experience any severe short-term or long-term complications, and all observed indicators have shown significant improvements, which is sufficient to demonstrate that the treatment method used in this study is safe and effective. That is, "Skull-femoral traction followed by osteotomy correction is a safe and effective treatment for severe scoliosis with split cord malformation."

7.In the discussion section, we have conducted a detailed analysis of the possible reasons for the improvement in pulmonary function and nutritional status. We believe that traction treatment reduces spinal rotation, alleviates rib rotation, indirectly increases thoracic cavity volume, thereby improving pulmonary function. It also reduces abdominal pressure and promotes gastrointestinal function. We consider these to be the possible physiological mechanisms, but further rigorous scientific experiments are needed to confirm them. Thank you for raising such rigorous scientific questions.

8.We have discussed the limitations of this study in the discussion section, of course, these are discussions regarding the treatment methods. Thank you to the reviewers for raising questions that made us aware of the shortcomings of our study. Therefore, in accordance with your valuable suggestions, we have added a supplement on the insufficient sample size of this study and measures for improvement at the end of the discussion.

Correction: the revised Introduction section is as follows. 

Introduction

Severe scoliosis accompanied with split cord malformation, a rare congenital spinal cord malformation, is structurally complex, with high surgical difficulty and risk. It poses a challenge in the treatment of scoliosis, particularly in cases of Pang I-type split cord malformation, where the preventive excision of the septum before spinal correction is a focus of debate in spinal surgery[1-4]. Split cord malformation (SCM), also known as diastematomyelia, is an infrequent variant of a closed neural tube defect characterized by the longitudinal division of the spinal cord, resulting in the formation of duplicated neural tubes or two hemicords within a single dural sleeve. This condition typically arises from anomalies within the spinal canal and congenital spinal deformities during embryogenesis, and is frequently comorbid with congenital scoliosis (CS)[5]. Studies by McMaster[6] and Bradford [7] have respectively indicated a 16% and 17% incidence of diastematomyelia among individuals with CS. Shen [8] identified SCM as the most prevalent spinal intradural anomaly in the context of CS. The occurrence of CS in SCM patients is documented to range from 52% to 79%[9-12]. Select the treatment plan based on the benefits and risks of alternative options[13]. Longitudinal spinal traction has gained consensus for improving flexibility and reducing correction risk in severe scoliosis[14-17]. Some scholars have reported favorable correction outcomes with two-stage correction and internal fixation following heavy traction for severe scoliosis[18-25]. It's believed that skull-femoral traction can effectively prevent excessive correction angles during traction, thereby avoiding spinal cord and important neurovascular damage[26,27]. In our department, skull-femoral traction followed by correction and internal fixation has yielded good results, with 82 cases completed, including 10 cases of accompanying split cord malformation, all without irreversible nerve damage. Therefore, a retrospective analysis was conducted on cases of severe scoliosis with split cord type I malformation treated with skull-femoral traction followed by two-stage correction, aiming to explore the safety and clinical efficacy of this treatment approach.

Discussion

Spinal cord split is divided into three types [28,29]: Type I is a double-tube type, where both halves of the spinal cord have their own independent dural tubes, separated by a bony or cartilaginous septum. For patients with congenital spinal deformities associated with spinal cord split without neurological symptoms or with stable neurological symptoms, especially for Type I spinal cord split patients, there is still controversy regarding the need for prophylactic resection of the spinal cord split before spinal correction surgery. Some scholars believe that significant changes occur in the morphology of the spinal canal during the reconstruction of the coronal and sagittal plane, causing a certain amount of traction and compression on the spinal cord, thereby exacerbating existing nerve damage or causing new nerve damage. They suggest that all patients with bony spinal cord split should undergo prophylactic laminectomy before spinal correction surgery [30]. However, Miller et al. [31] reported that among 33 patients who underwent laminectomy for spinal cord split, 22 patients did not show significant improvement in postoperative neurological symptoms, and one patient experienced worsened neurological symptoms. They believe that surgical intervention for spinal cord split patients should be approached with extreme caution. Chen et al. [4] also found that the incidence of complications in patients with congenital spinal deformities associated with spinal cord split who underwent one-stage posterior spinal osteotomy correction surgery was not significantly higher compared to patients without spinal cord split. Shen et al.'s [32] prospective study results indicated that for patients with spinal cord split who have no neurological symptoms or stable neurological symptoms, a simple spinal correction surgery is safe and effective without the need for prophylactic laminectomy. Zhu et al. [33] suggested that for patients with severe spinal deformities and Type I spinal cord split who require three-column osteotomy during surgery, if the bony split is within the osteotomy level, it is advisable to simultaneously remove the lamina during the correction surgery. The viewpoints of the aforementioned authors are based on one-stage surgery and do not consider the impact of traction on spinal cord split. Multiple perspectives suggest that longitudinal spinal traction has a positive effect on improving flexibility and reducing the risk of correction in severe rigid spinal scoliosis, but there is no clear study on the impact of traction therapy on spinal cord split.

This article aims to analyze the safety and effectiveness of cranial traction on severe spinal scoliosis accompanied by spinal cord bony split. Based on clinical practice, the author summarizes the following points after performing cranial femoral traction followed by one-stage surgery on suitable cases: 1. Cranial femoral traction has strong stability, with continuous and stable traction force. Its effect on longitudinal spinal traction can be seen as a tearing-repair reconstruction effect on contracted soft tissues. Observations on patient's skin appearance, skin tension, and muscle tension lead to the conclusion that it has similar effects on spinal intervertebral discs, facet joints, ligaments, and other tissues. 2. Cranial femoral traction has good consistency in traction force direction, restoring coronal and sagittal plane balance of the spine. Observations of severe spinal scoliosis patients after traction lead to this conclusion. 3. Post-cranial femoral traction increases thoracic cavity volume, improves lung function indicators to varying degrees, alleviates abdominal organ compression, etc., leading to improvements in patient nutritional status, enhancing perioperative safety, and surgical tolerance. Mesenteric vessels, mesenteric ligaments, etc., undergo reconstruction during traction, reducing postoperative complication risks. 4. Post-cranial femoral traction reduces the angle of lateral scoliosis, significantly influencing surgical decisions. The main curve, compensatory curve, scoliosis, and rotation all show varying degrees of improvement, with some correction rates reaching 50%. Compared to one-stage correction, the decision-making in traction surgery regarding osteotomy level, osteotomy range, fusion range all decreases, reducing intraoperative correction difficulty, surgical time, and blood loss significantly. 5. Compared to gravity traction and halo pelvic traction, cranial femoral traction is stable, continuous, and exerts strong traction force, allowing patients to choose rest, walk freely, and take care of themselves, thus enhancing their dignity. 6. Cranial femoral traction can increase the tolerance of the spinal cord to deformation; some patients experience an 8 cm increase in height after traction, while the total length of the spinal canal remains unchanged due to changes in spinal cord morphology with the spinal canal. Blood vessels and spinal cord adapt during slow deformation, stabilizing in the final traction state. Cranial femoral traction guides the degree of correction in surgery. 7. For those with abnormalities in the spinal canal or spinal cord, progressive traction allows observation of neurological function status. If any neurological abnormalities appear, traction is immediately relaxed, leading to neurological function recovery, indicating the limit of correction and avoiding over-correction in surgery. Currently used traction methods include Halo-pelvic traction, Halo-gravity traction, and Halo-femoral trochanter traction. Qiu et al. [34,35] reported that Halo wheelchair suspension gravity traction can improve the correction effect of spinal scoliosis, especially idiopathic scoliosis. However, gravity traction requires a longer duration, with reported maximum traction times in China ranging from 4 to 15 weeks, averaging 10.4 weeks, leading to prolonged hospital stays. Moreover, the stability of such traction is po

---

## [Editor Report · Decision Letter 1]

12 Aug 2024

PONE-D-24-23086R1Skull-femoral traction followed by osteotomy correction is a safe and effective treatment for severe scoliosis with split cord malformationPLOS ONE

Dear Dr. Guo,

Thank you for submitting your manuscript to PLOS ONE. After careful consideration, we feel that it has merit but does not fully meet PLOS ONE’s publication criteria as it currently stands. Therefore, we invite you to submit a revised version of the manuscript that addresses the points raised during the review process.

We look forward to receiving your revised manuscript.

Kind regards,

Barry Kweh

Academic Editor

PLOS ONE

Journal Requirements:

Additional Editor Comments:

A further discussion of the long-term follow-up and medical complications of the disease would improve the manuscript.

---

## [Author Response · Author response to Decision Letter 1]

12 Aug 2024

13-Aug-2024

Manuscript ID: PONE-D-24-23086

Dear Prof. Barry Kweh，

We hereby re-submit our revised manuscript entitled “Skull-femoral traction followed by osteotomy correction is a safe and effective treatment for severe scoliosis with split cord malformation” (Manuscript ID: PONE-D-24-23086). 

The author's team is honored to hear from the editor and receive thoughtful comments and feedback from the reviewers. We appreciate your valuable advice and have made substantial revisions to our manuscript accordingly. 

1.We have carefully compared our manuscript with the template format of your journal and have modified the format of the manuscript to conform as closely as possible to the requirements of your publication.

2.We have supplemented all the raw data required for the research findings in the revised submission files, which have been uploaded as supporting information.

3.All of our authors are in agreement to share our raw data.

4.The corresponding author has completed the ORCID authentication.

5.An ethical statement has been added in the Methods section, explicitly stating that the patient has given consent for the publication of case details.

6.We have added a detailed discussion on the long-term follow-up and medical complications of the disease in the discussion section.

We have provided point-to-point responses to all the reviewer's questions in the uploaded file “Response to Reviewers” and made corresponding modifications to the manuscript based on the questions raised by the reviewers. All the revisions we have made with the manuscript are marked in red in the uploaded file “Revised Manuscript with Track Changes”. The updated version of supporting information is in the file named “revised supporting information”.

We hope our work and effort could move this article closer for publication in your esteemed journal. If you have any additional question or concern, please do not hesitate to contact me at your convenience. 

Sincerely, 

Tao Guo

---

## [Editor Report · Decision Letter 2]

16 Aug 2024

Skull-femoral traction followed by osteotomy correction is a safe and effective treatment for severe scoliosis with split cord malformation

PONE-D-24-23086R2

Dear Dr. Guo,

We’re pleased to inform you that your manuscript has been judged scientifically suitable for publication and will be formally accepted for publication once it meets all outstanding technical requirements.

Kind regards,

Barry Kweh

Academic Editor

PLOS ONE

Additional Editor Comments (optional):

A well written article which has justified statistical methods and significantly improved the depth as well as breadth of the discussion.
---

## [Editor Report · Acceptance letter]

3 Sep 2024

PONE-D-24-23086R2 

PLOS ONE

Dear Dr. Guo, 

I'm pleased to inform you that your manuscript has been deemed suitable for publication in PLOS ONE. Congratulations! Your manuscript is now being handed over to our production team.

Kind regards, 

on behalf of

Dr. Barry Kweh 

Academic Editor

PLOS ONE